ClBRN1 from Chrysanthemum lavandulifolium enhances the stress resistance of transgenic Arabidopsis

Li Yanxi
He Wenting
Liu Yueyue
Mei Chendi
Wang Hai wh672831@126.com
Song Xuebin xuebinsong@sina.cn
College of Landscape Architecture and Forestry, Qingdao Agricultural University , Qingdao, Shandong , China
Abd El-Moneim Diaa
Electronic publication date: 2024 Dec 12
Publication date: 2024
Volume: 12
Electronic Location ID: e18620
Received 2024 Aug 28; Accepted 2024 Nov 11
Copyright: © 2024 Li et al.
Copyright year: 2024
Copyright holder: Li et al.
License: This is an open access article distributed under the terms of the Creative Commons Attribution License, which permits unrestricted use, distribution, reproduction and adaptation in any medium and for any purpose provided that it is properly attributed. For attribution, the original author(s), title, publication source (PeerJ) and either DOI or URL of the article must be cited.
License URL: https://creativecommons.org/licenses/by/4.0/

Keywords: Chrysanthemum lavandulifolium, ClBRN1, Stress resistance, Functional verification

Funding: National Natural Science Foundation of China 32101580 Forestry and Grassland Germplasm Resources Center of Shandong Province 6602423134 Doctoral Foundation of Qingdao Agricultural University 6631120091 and 6631122019 This work was supported by the National Natural Science Foundation of China (32101580), the Herbaceous Germplasm Resources Survey and Collection Project of Forestry and Grassland Germplasm Resources Center of Shandong Province (6602423134), and the Doctoral Foundation of Qingdao Agricultural University (6631120091, 6631122019). The funders had no role in study design, data collection and analysis, decision to publish, or preparation of the manuscript.

==============================
Background

Chrysanthemum (Chrysanthemum×morifolium Ramat.) is a particularly important autumn perennial flower for potted plant, flower bed and border, and cut flower with high ornamental value. However, abiotic stress can affect the ornamental quality of Chrysanthemum. NAC (NAM, ATAF1-2, and CUC2) transcription factors (TFs) play an important role in regulating plant growth and development, as well as responding to abiotic stresses.

Methods

In this study, the ClBRN1 (Chrysanthemum lavandulifolium BEARSKIN gene) was isolated from the Chrysanthemum model plant C. lavandulifolium. And analyze the function of the gene through bioinformatics, subcellular localization and overexpression.

Results

Bioinformatics analysis showed that the ClBRN1 gene was a member of the NAC TFs family, with a CDS (coding sequence) length of 1,080 bp and encoding 359 amino acids. The subcellular localization results found that the gene was located in the nucleus and cell membrane. Furthermore, the transgenic results in Arabidopsis thaliana showed that the gene significantly reduces plant height while improving salt and low temperature tolerance. Observation of paraffin sections of Arabidopsis stems also revealed that the secondary cell wall of overexpressing Arabidopsis stems was significantly thicker than that of wild-type. The above results indicate that the ClBRN1 gene may play an important role in regulating plant resistance to abiotic stress. This study will provide new insights for molecular breeding of resistant chrysanthemums in the future.

Introduction

Chrysanthemum (Chrysanthemum×morifolium Ramat.) is an particularly important autumn and perennial flower for potted plant, flower bed and border, and cut flower with high ornamental value (Zhang, 2023). However, Chrysanthemum faces various types of abiotic stresses during their growth process. This will affect the growth and development of chrysanthemums, and leading to a decrease in ornamental quality. This is a major challenge for Chrysanthemum breeding. Molecular breeding technology can selectively cultivate new varieties with high quality, resistance, and ornamental value (Silan et al., 2014). At present, due to the polyploidy and complex genetic background of the Chrysanthemum varieties, their molecular breeding work is difficult to carry out. Most notably, Chrysanthemum lavandulifolium, a perennial herb in the Asteraceae family, is one of the important parents of modern Chrysanthemum. It has the advantages of fast growth rate and high fruiting rate, and is often used as a model plant for Chrysanthemum research due to its diploid nature. In addition, it has strong resistance and is suitable for mining resistance genes (Wang, 2021). The research on resistance breeding in C. lavandulifolium can provide scientific basis for breeding high resistance varieties of Chrysanthemum and reducing the loss in the process of Chrysanthemum application. Therefore, this study used C. lavandulifolium as the experimental material to investigate the stress resistance of Chrysanthemum.

Transcription factor genes in plants play important roles in regulating plant growth, development, and stress response (Dong, 2014). There are 58 transcription factor families in known plants, such as the MYB, NAC, and WRKY transcription factor families, which are the most studied families (Ng, Abeysinghe & Kamali, 2018). Among them, NAC (NAM, ATAF1-2, and CUC2) is a plant-specific transcription factor. It is composed of a conserved N-terminal protein binding domain (PBD) and a variable C-terminal transcription regulatory region (TRR). NAC transcription factors play a crucial role in plant growth and development, especially in response to biotic and abiotic stresses (Wei et al., 2008). In growth and development, NAC transcription factors involved in the regulation of secondary cell wall development, root development, leaf senescence, flower organ formation and fruit maturation (Giovannoni, 2004; Guo et al., 2018; Hao et al., 2011; Kim, Nam & Lim, 2016; Zhong, Demura & Ye, 2006). Research has found that NAC transcription factors play an important regulatory role in the stress of ornamental plants, mainly focusing on the study of their abiotic stress such as drought, salt, alkali, cold, and heat (Deng et al., 2023).

The OsNAC7 subfamily is the most studied subfamily in the NAC family and includes SND1, NST1, URP7, BRN1/2, and VND1/2/3/4/5/6/7 genes. The main functions of this subfamily are regulating the formation of the secondary cell wall (SCW) in stems, roots and anthers (Bennett et al., 2010; Mitsuda et al., 2007; Yamaguchi et al., 2010; Zhong, Demura & Ye, 2006). There are complex interactions between the NAC transcription factor subgroup OsNAC7 and SCW genes (Lee et al., 2019; Mitsuda et al., 2007; Tan et al., 2018; Zhong et al., 2021). The URP7, BRN2 and BRN1 genes are involved mainly in root cap development and control root cap maturation in A. thaliana (Bennett et al., 2010; Fendrych et al., 2014). In A. thaliana, BRN2 and SMB regulate root cap maturation in a partially redundant manner, and BRN1 and BRN2 control the maturation process of the cell wall, which is necessary for the separation of the root cap from the root (Bennett et al., 2010). Kamiya et al. (2016) found that the expression of BRN1 and BRN2 is closely related to the location of cells on the root surface. At present, most studies on BRN1 gene have been conducted in model plants such as A. thaliana, but few studies have been conducted in Chrysanthemum.

As a complex phenolic polymer, lignin widely involved in various aspects of plant growth, development and stress response. The changes of lignin content can directly affect plant growth and development, as well as biotic and abiotic resistance (Bonawitz et al., 2014). For example, lignin is closely related to stem strength, growth amount, flowering time, disease resistance, insect resistance, salt and alkali resistance, waterlogging resistance and cold resistance of plants (Li, 2003). Overexpression of BplMYB46 gene in Betula platyphylla can improve the tolerance of transgenic birch to salt stress and osmotic stress, and enhance the deposition of lignin (Guo et al., 2017). MdSND1 is directly involved in the regulation of lignin biosynthesis and the signal transduction pathway involved in the response to both salt and osmotic stress in apple (Malus × domestica Borkh.) (Chen et al., 2020). These results suggest that there may be lignin biosynthesis pathways involved in the molecular regulatory mechanisms affecting plant abiotic stress. Most notably, the OsNAC7 subfamily can regulate lignin synthesis, and BRN1 is one of the important members of this family. Furthermore, it is speculated that the BRN1 gene may be involved in lignin biosynthesis and plant abiotic stress. At present, research on the BRN1 gene mainly focuses on root cap mucus secretion and root cap maturation, with little research on abiotic stress (Bennett et al., 2010; Liu et al., 2024). Therefore, this study took BRN1 as a research object to explore its role in regulating stress resistance.

Stress resistance is one of the important goals of Chrysanthemum breeding, and molecular breeding technology is an important breeding method. In this study, ClBRN1 gene was cloned. And then, the gene function was analyzed through bioinformatics analysis, paraffin sectioning, subcellular localization, and overexpression methods. At the same time, stresses treatment was applied to overexpressing and wild-type plants to further validate the resistance of the gene to harsh environments. The aim of this study is to explore the function of ClBRN1 gene in response to stress that can be used for transgenic breeding to enhance the resistance in Chrysanthemum.

Materials & methods

Plant materials and growing environment

The materials used in this experiment were C. lavandulifolium, Nicotiana benthamiana and A. thaliana. The tissue seedlings of C. lavandulifolium were obtained from the School of Landscape and Forestry of Qingdao Agricultural University (Qingdao, China) and were cultured in 1/2 MS media. A. thaliana seeds were seeded on 1/2 MS medium, germinated 2–3 days later, transplanted to 1–3 mm peat at 2–3 true leaf stage, and cultured in the artificial climate chamber of Qingdao Agricultural University (Qingdao, China) (temperature: ± 25 °C; relative humidity: 60%; light intensity: 2,500 lx; illumination time: 16 h; darkness: 8 h). N. benthamiana was seeded on MS medium under the same culture conditions as A. thaliana.

Strains and vectors

The main strains used in this study were Escherichia coli DH5α; Agrobacterium tumefaciens GV3101. The above strains were purchased from Shanghai Weidi Biotechnology Co., Ltd., Shanghai, China. The overexpressed Super1300-GFP were all retained in the laboratory.

Reagent

2×Phanta Max Master Mix, 2×TaqPlus Master Mix, HiScript III RT SuperMix for qPCR (+gDNA wiper), FastPure Universal Plant Total RNA Isolation Kit, ClonExpress Ultra One Step Cloning Kit, TOPO vectors (5 min TA/Blunt-Zero Cloning Kit) purchased from Nanjing Vazyme Biotechnology Co. Ltd., Nanjing, Jiangsu, China. Kanamycin, Rifampicin, Timentin, Gentamycin Sulfate, Hygromycin, MS and LB medium purchased from Beijing Solarbio Science & Technology Co., Ltd., Beijing, China; Restriction endonuclease PstI and KpnI-HF® purchased from New England Biolabs (Beijing) Co. Ltd., Beijing, China.

Total RNA extraction and cDNA synthesis

Total RNA was extracted from fresh leaves of C. lavandulifolium. Total RNA extraction was performed according to the procedure of the FastPure Universal Plant Total RNA Isolation Kit (Vazyme, Jiangsu, China); first-strand cDNA was synthesized by HiScript III RT SuperMix for qPCR (+gDNA wiper) (Vazyme, Jiangsu, China).

Target gene cloning

Using cDNA as template, PCR reaction was performed with primers in Table 1. The PCR amplification system was 25 µL and the reaction system was as follows: 2 × Phanta Max Master Mix: 12.5 µL; ClBRN1-F: 1 µL; ClBRN1-R: 1 µL; cDNA: 2 µL; ddH2O: 8.5 µL. The reaction procedure was predenaturation at 95 °C for 3 min. Denaturation at 95 °C for 15 s, annealing at 56 °C for 15 s, extension at 72 °C for 60 s, 35 cycles; Extend at 72 °C for 5 min; The PCR products were separated by agarose gel electrophoresis, and the PCR products were recovered by bands with the same length as the target gene. cDNA clones were connected to TOPO vector (Vazyme, Jiangsu, China). The link product transformed Escherichia Coli DH5α (WEIDI, Guangdong, China). Positive clones were obtained after LB medium containing 50 mg/L (Kanamycin) was screened and sent to Sangong BioEngineering (Shanghai, China) Co., Ltd. for sequencing.

Table 1 Primers’ names and sequences.

Primer name	Sequence (5′→3′)	TM/°C	
ClBRN1 cloning primer	
ClBRN1-F	ATGGGATCATCGAATGGTGG	54.69	
ClBRN1-R	CTATTTACCGTACCCCCAAAAGT	54.20	
ClBRN1 overexpression primer	
ClBRN1-Pst I	gctctagaatgggatcatcgaatggtgg	61.00	
ClBRN1-Kpn I	aactgcagtttaccgtacccccaaaa	60.71	

Bioinformatics analysis of sequences

Sequences were compared by DNAMAN (version 8.0, Lynnon Biosoft, Quebec, QC, Canada); Evolutionary analysis and homology analysis were performed by MEGA (version 7.0, Mega Limited, Auckland, New Zealand).

Construction of the overexpression vector

According to the ORF region of the ClBRN1 gene and the multiple cloning sites of the Super1300::GFP vector, the PstI and KpnI restriction enzyme sites were selected to design the target fragment primers (Table 1). The target fragments containing the endo nuclide sites and without stop codon were obtained by PCR reaction and agarose gel recovery. The target fragment was inserted into the overexpression vector Super1300, and the recombinant plasmid was transformed into DH5α to obtain a positive clone. After sequencing, GV3101 was transformed for infection.

Observation of subcellular localization signals

The constructed Super1300::ClBRN1::GFP overexpression vector was transiently transformed into N. benthamiana via A. tumefaciens GV3101 (Weidi, Guangdong, China). The infection solution was injected into the backs of 4- to 6-week-old (Hu et al., 2022) the tobacco leaves. After one day of culture in the dark, the N. benthamiana epidermal cells were removed, and the GFP (green fluorescent protein) signal was observed by laser confocal microscopy (TCSsp5II03040101; Agilent, Santa Clara, CA, USA). The Super1300::GFP empty vector was used as a control.

Genetic transformation and screening of Arabidopsis

The Super1300::ClBRN1::GFP overexpression vector was subsequently transformed into A. thaliana by the Agrobacterium-mediated floral dip method (Wenwen et al., 2014). The transgenic A. thaliana seeds were screened by resistance screening medium (1/2 MS + 30 g/L sucrose + 6 g/L agarose + 50 mg/L hygromycin), and homozygous offspring were obtained from the T2 generation.

Stress treatment and phenotypic observation

The A. thaliana transgenic plants and wild type at the seedling stage and the mature stage were treated with salt and low temperatures. At the seedling stage, the transgenic A. thaliana seedlings and wild type were treated with abiotic stress, and the A. thaliana seedlings with the same growth rate were selected and transferred to the treatment medium, and the 1/2 MS medium was selected as the basic medium. Approximately 200 mmol/L was used for salt treatment and the basal medium was used for the 4 °C low-temperature treatment and control. After 3 days of treatment, A. thaliana was photographed and recorded. A. thaliana was removed from the medium, cleaned with running water, and the surface was dried with filter paper, and the phenotype was statistically recorded. Also, stress treatments were applied to the A. thaliana at the mature stage. A. thaliana with similar growth state was selected for treatment. Salt treatment was simulated by applying 200 mmol/L NaCl solution. The samples were transferred to the low-temperature treatment using a low temperature incubator at 4 °C. It was processed for a total of 12 days, during which constant photography was taken.

Paraffin sections of Arabidopsis stems

Paraffin cross-sections of A. thaliana stems were made. First, stems 3–5 cm at the base of the main stem were cut 40–50 days after the growth of A. thaliana, and the excess leaves on the stem were removed and fixed in FAA fixative for 24 h. Dehydration was staged with different concentrations of alcohol. A. thaliana was transparent by mixing ethanol with xylene. Wax was dipped in EP tube mixed with xylene and liquid paraffin. Then A. thaliana was embedded by embedding machine. After embedding, it was sliced and dewaxed, and then stained with toluidine blue. Finally, the paraffin sections of the A. thaliana stems were placed on a Zeiss microscope (Axio Scope A103040108; Zeiss, Oberkochen, Baden-Württemberg, Germany) to observe cross section of stems of different A. thaliana.

Statistical analysis

Three biological replicates and technical replicates were performed for treatments in this study. Excel2010 was used to organize the data, SPSS27 was used for statistical analysis, one-way ANOVA and least significant difference method were used to analyze the data (P < 0.05), and Origin2022 software was used to plot the data.

Results

ClBRN1 gene cloning and bioinformatics analysis

In this study, the ClBRN1 gene was cloned from C. lavandulifolium. The CDS of the ClBRN1 gene is 1,080 bp in length and encodes 359 amino acids. The nucleotide sequence of the ClBRN1 gene was compared with the nucleotide sequence of Cn0053200 of Chrysanthemum nankingense in the chrysanthemum genome database, which revealed that the two gene sequences were essentially the same (Fig. 1A). Conserved domain analysis of the ClBRN1 gene (Fig. 1B) showed that the gene contained the NAM conserved domain and was a member of the NAC transcription factor family. Phylogenetic tree analysis (Fig. 1C) showed that the gene was in a branch with Artemisia annua and Tanacetum cinerariifolium, and the genetic relationship was close. Phylogenetic analysis of the ClBRN1 protein with 12 members of the A. thaliana OsNAC7 subfamily (Fig. 1D) showed that the BRN1 protein of C. lavandulifolium had the closest relationship with BRN1 and BRN2 of A. thaliana, suggesting that the BRN1 protein of C. lavandulifolium protein might be a BRN1 protein.

Figure 1 Bioinformatic analysis of ClBRN1.

(A) Cloning and comparative analysis of the ClBRN1 gene in C. lavandulifolium. (B) Analysis of the conserved domain of the ClBRN1 gene in C. lavandulifolium. (C) Evolutionary analysis of the BRN1 protein in C. lavandulifolium. (D) Homology comparison between the BRN1 protein in C. lavandulifolium and members of the A. thaliana OsNAC7 subfamily.

Expression localization of the ClBRN1 gene

To analyze the localization of the protein encoded by the ClBRN1 gene in cells, the Super1300::ClBRN1::GFP vector was constructed and transiently transformed into N. benthamiana. As shown in Fig. 2, the fluorescence of Super1300::ClBRN1::GFP mainly appeared in the cell membrane and nucleus of epidermal cells, while the GFP signal of the control Super1300::GFP empty vector mainly appeared around the cell membrane, cytoplasm and nucleus. These results indicate that the BRN1 protein of C. lavandulifolium is localized to the cell membrane and nucleus.

Figure 2 Subcellular localization analysis of ClBRN1.

Effects of ClBRN1 gene overexpression on the growth and development of Arabidopsis

In order to investigate the effect of ClBRN1 on growth and development, the phenotypes of transgenic A. thaliana and wild type were observed in this study. Their rosette diameter, plant height and flowering time were recorded. The results (Fig. 3A) showed that there was little difference in rosette diameter between transgenic A. thaliana and wild type. The statistical results of A. thaliana height (Fig. 3B) showed that there was no significant difference in height between transgenic A. thaliana and wild type before day 17. After 17 days, the height of transgenic A. thaliana began to be significantly lower than that of wild type, decreasing by about 37%. With respect to the growth of A. thaliana inflorescence, the inflorescence elongation rate and height of wild-type A. thaliana were greater than those of transgenic Arabidopsis. Wild-type A. thaliana began to grow inflorescences 9–12 days after transplanting and began to bloom on the 12th day. The inflorescences of ClBRN1 transgenic A. thaliana began to grow inflorescences 10–14 days after transplanting and began to bloom on the 12th day. These results indicated that the plant height of transgenic A. thaliana plants was inhibited.

Figure 3 Growth status of Arabidopsis.

(A) Rosette diameter statistics of A. thaliana. (B) Growth height statistics of A. thaliana. The same letters are not significantly different. The different letters are significantly different (P < 0.05). Bars indicate standard errors (n = 3). WT, wild type; OE-ClBRN1, overexpression-ClBRN1.

Response of Arabidopsis to 4 °C low-temperature and NaCl treatments

In order to explore the function of this gene in regulating plant stress resistance, A. thaliana at seedling stage and mature stage were treated with stress respectively, and wild type was used as control. The results showed that at the seedling stage, after 3 days of treatment at 4 °C (Fig. 4A), A. thaliana overexpressed with ClBRN1 produced more short lateral roots than the control group. The root elongation, lateral root length, lateral root number and survival rate of A. thaliana were statistically analyzed (Fig. 5). At 4 °C, the root elongation of transgenic A. thaliana was 39% higher than that of control, the lateral root length was 73% higher than that of wild type, and the lateral root number was 42% higher than that of wild type, with significant differences (P < 0.05). There was no significant difference in survival rate. After 3 days of 200 mmol/L NaCl treatment, there was no significant difference in root elongation compared with the control group, but the number of lateral roots was 33% higher than that of the wild type, and the survival rate was 25% higher than that of the wild type. At the mature stage, the phenotypes of overexpressed A. thaliana were superior to those of the wild type at low temperature (4 °C) (Fig. 4B). The phenotype of overexpressed A. thaliana was also superior to that of the wild type under NaCl treatment (Fig. 4C). These results indicate that overexpression of ClBRN1 gene can promote the generation of transgenic A. thaliana lateral roots and show strong resistance under low temperature and salt stress. ClBRN1 gene increased the tolerance of transgenic A. thaliana to NaCl and low temperature stress.

Figure 4 Phenotype of Arabidopsis under 4 °C and NaCl treatments.

(A) Growth of A. thaliana at 4 °C and NaCl treatment for 3 days at the seedling stage. (B) Growth of A. thaliana at 4 °C at the mature stage. (C) Growth of A. thaliana in the 200 mmol/L NaCl treatment group at the mature stage. WT, wild type; OE-ClBRN1, overexpression-ClBRN1.

Figure 5 Growth status statistics of Arabidopsis at 4 °C and NaCl treatments for 3 days.

(A) Root elongation. (B) Mean lateral root length. (C) Lateral root number. (D) Survival rate. The same letters are not significantly different. The different letters are significantly different (P < 0.05). Bars indicate standard errors (n = 3). WT, wild type; OE-ClBRN1, overexpression-ClBRN1.

Paraffin section observation of Arabidopsis stems

The OsNAC7 subfamily regulates lignin synthesis and secondary cell wall formation. Therefore, the paraffin sections of A. thaliana stems were prepared and observed. Three- to five-centimeter-long stems of transgenic A. thaliana and wild-type plants were selected for paraffin section observation. The cell wall thickness of secondary xylem cells in super transgenic A. thaliana plants was much larger than that of wild-type plants (Figs. 6A and 6C) (P < 0.05). The paraffin sections of transgenic and control A. thaliana were obtained and observed, and the cell wall thickness of secondary xylem cells was measured (Figs. 6B and 6D). The cell wall of wild-type A. thaliana was measured, and the average thickness was 3.79 μm. The cell wall of transgenic A. thaliana plants was measured, and the average thickness was 8.19 μm. Compared with that of the control group, the cell wall thickness of the secondary xylem cells at the base of ClBRN1 transgenic A. thaliana increased by 54%, and the cell wall thickness of the secondary xylem increased significantly (Fig. 6E). These results indicate that the ClBRN1 gene regulates the secondary cell wall and thickens it.

Figure 6 Paraffin section observation of Arabidopsis stems.

(A and B) Cross-sectional paraffin sections of wild-type A. thaliana; (C and D) Transgenic A. thaliana stem cross-sectional paraffin sections; (E) secondary cell wall thickness of A. thaliana stems. The same letters are not significantly different. The different letters are significantly different (P < 0.05). Bars indicate standard errors (n = 3). WT, wild type; OE-ClBRN1, overexpression-ClBRN1.

Discussion

NAC is a unique transcription factor in higher plants and one of the largest transcription factor families in plant genomes (Wei et al., 2008). NAC transcription factors have a common conserved region at the N-terminus called the NAC domain, which is unique to NAC transcription factors. NAC transcription factors play important roles in plant growth and development, organ formation, hormone regulation and stress response. These transcription factors are a hot topic in the study of plant gene function and gene expression regulation (Olsen et al., 2005; Zheng et al., 2009). Due to the polyploidy and complex genetic background of the Chrysanthemum varieties, their molecular breeding work is difficult to carry out. And, few studies have investigated the OsNAC7 gene of Chrysanthemum. C. lavandulifolium, a perennial herb in the Asteraceae family, is one of the important parents of modern Chrysanthemum. It has the advantages of fast growth rate and high fruiting rate, and is often used as a model plant for Chrysanthemum research due to its diploid nature. Therefore, this study cloned the ClBRN1 gene of the OsNAC7 gene family from C. lavandulifolium. Conserved domain analysis showed that the gene contained a NAC transcription factor family-specific NAM conserved domain. Phylogenetic analysis showed that the protein was closely related to BRN1 and BRN2 of the A. thaliana OsNAC7 subfamily, and it was speculated that the gene was ClBRN1.NAC transcription factors can slow plant growth. Saad et al. (2013) found that the plant height of rice seedlings with SNAC1 gene was shorter than that of the control group. Allogeneic expression of EsNAC1 (Eutrema salsugineum) delayed the vegetative growth of A. thaliana (Liu et al., 2018). The growth of plants is inhibited by overexpressing OsNAC6 gene (Takasaki et al., 2010). This study found that overexpression of the ClBRN1 gene inhibited plant height of A. thaliana This is basically consistent with the above research results. Further analysis of the above results reveals that the NAC gene may alter the life cycle of plants by regulating cell division. Membrane proteolysis and cytokinin signal can activate the expression of NTM1 (NAC with transmembrane motif1), which induces a series of CDK (cyclin-de pendent kinases) suppressor gene expression in NTM1 mutants. These genes inhibit the synthesis of histone H4, which inhibits cell division and leads to delayed growth (Kim et al., 2007). Therefore, it is speculated that ClBRN1 is involved in the regulation of cell division to inhibit inhibiting plant growth.

Chrysanthemums are a variety of traditional Chinese ornamental flowers, and their growth and ornamental value can be affected under stress conditions. Abiotic stress can affect root morphology and inhibit normal plant growth. Roots play an important role in coping with abiotic stress, and strong lateral roots help plants absorb more water and minerals under stress. As is well known, plants with strong stress resistance have larger root systems and stronger absorption capacity (Coleman, 2007). Under stress treatment, overexpression of MsLEA4-4 plants (Medicago sativa L.) resulted in more lateral roots, higher chlorophyll content, and higher survival rate compared to the wild type (Jia et al., 2020). NAC gene expression can promote lateral root development. Under stress, NAC gene is induced to mediate auxin signal transduction to promote lateral root development. This transcription factor can activate the expression of downstream auxin response genes, and overexpression can promote lateral root development (He et al., 2005). Seven cotton genes, GhNAC7-GhNAC13 (Gossypium hirsutum), are preferentially expressed in roots and respond to abiotic stress during root development (Huang et al., 2013). Similarly, this study found that ClBRN1-overexpressing A. thaliana had more lateral roots, thereby enhancing salt and low temperature tolerance.

Previous studies have found that secondary cell walls can participate in both biotic and abiotic stress responses in plants. Lignin is an important component of the secondary wall of plant cells, and its content and expression of related enzymes are important factors affecting plant stress resistance. NAC transcription factor genes can act on the upstream of plant secondary wall synthesis regulatory network and play an important role in lignin synthesis (Wang, 2021). The biosynthesis of the plant secondary cell wall is positively regulated by NAC transcription factors (Zhong & Ye, 2014). The overexpression of SMB, BRN1 and BRN2 in A. thaliana induced the formation of secondary cell walls in the root vascular system, which was consistent with the overexpression of VND/NST/SND family members (Kamiya et al., 2016). SND1 simultaneously promotes lignin synthesis and maintains ABA concentration (Jeong et al., 2018). NST1 can positively regulate the formation of secondary cell walls and lignin deposition in Arabidopsis thaliana (Liu et al., 2021). And, BpNAC012 (Betula platyphylla) can also positively regulate abiotic stress response and secondary wall biosynthesis (Hu, Zhang & Yang, 2019). In this study, it was found that the ClBRN1 gene increased the thickness of secondary cell walls in Arabidopsis, indicating that it may be involved in regulating the growth of secondary cell walls. Therefore, it is speculated that the ClBRN1 gene can enhances stress resistance in Arabidopsis by positively regulating the thickening of secondary cell walls. However, further verification is needed regarding the relationship between secondary cell wall thickening and enhanced resistance.

Conclusions

In summary, ClBRN1 was cloned from C. lavandulifolium and identified as a gene of the OsNAC7 subfamily. Overexpression of ClBRN1 gene inhibited the plant height of transgenic A. thaliana, produce more lateral roots and also thickened the secondary cell wall to enhance resistance to low temperature and salt stress. This not only provides a new insight into the function of BRN1 gene, but also provides valuable information for the directional breeding of high quality and high resistance Chrysanthemum and other ornamental plant varieties. However, the ClBRN1 gene has only been studied in model plant A. thaliana, and will be further studied in C. lavandulifolium and Chrysanthemum. Which downstream genes of ClBRN1 are specifically regulated in stress response, and which metabolic pathways are involved in regulating plant growth and secondary cell wall formation are the future research directions.

Supplemental Information

Supplemental Information 1 Growth index statistics of Arabidopsis.

The data in Figures 3, 5, 6.

Additional Information and Declarations

Competing Interests

Author Contributions

DNA Deposition

Data Availability

The authors declare that they have no competing interests.

Yanxi Li conceived and designed the experiments, performed the experiments, analyzed the data, prepared figures and/or tables, and approved the final draft.

Wenting He conceived and designed the experiments, performed the experiments, analyzed the data, prepared figures and/or tables, and approved the final draft.

Yueyue Liu analyzed the data, prepared figures and/or tables, and approved the final draft.

Chendi Mei analyzed the data, prepared figures and/or tables, and approved the final draft.

Hai Wang conceived and designed the experiments, authored or reviewed drafts of the article, and approved the final draft.

Xuebin Song conceived and designed the experiments, authored or reviewed drafts of the article, and approved the final draft.

The following information was supplied regarding the deposition of DNA sequences:

The nucleotide sequence of Cn0053200 is available at the chrysanthemum genome database: http://210.22.121.250:8880/asteraceae/homePage.

The following information was supplied regarding data availability:

The raw measurements in Figures 3, 5, 6 are in available in the Supplemental Files.

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
