# Peer review of "ClBRN1 from Chrysanthemum lavandulifolium enhances the stress resistance of transgenic Arabidopsis"

_PeerJ, doi:10.7717/peerj.18620_

## Round 0.1 · original submission · Major Revisions

Dear Authors

The manuscript needs a revision before publication. The authors are invited to revise the paper considering all the suggestions made by the reviewers. Please note that the requested changes are required for publication.
With Thanks

Reviewer 1 ·

Basic reporting

The English of the article should be rechecked. Expressions suitable for academic writing should be included.

Up-to-date references were used.

The article structure and figures are appropriate, but there are spelling errors in the captions of figures and tables. They should be corrected. For example, Arabidopsis thaliana should have a lowercase “t” and the entire species name should be italicized. All species name like; Arabidopsis, Chrysanthemum or etc. should be italic.

If gene names are mentioned, italics should be used; if protein names are mentioned, plain text should be used. All text should be checked and corrected.

The typefaces of the abstract and the main text are different. they should be written in the same format and justified. This way it looks sloppy.

Experimental design

The article is good in terms of experimental phases and editing. The results are explained, but the discussion could be improved by linking the results to case studies.

Validity of the findings

The findings are well explained in terms of rationality, statistics and the article. However, how the results are interpreted and discussed needs to be strengthened.

Additional comments

Line 1. Lavandulifolium should start with lowercase "L"
Line 3, 32, 33, 35, 52, 57, 58, 72, 146, 167,..., 318 Arabidopsis should be italic (same with other species names)
What does the Y coordinate of the graph in part E of figure 6 refer to?
figure captions should be more descriptive.
The whole text should be checked again for grammar and spelling.

Reviewer 2 ·

Basic reporting

The study titled "ClBRN1 from Chrysanthemum lavandulifolium Enhances the Stress Resistance of Transgenic Arabidopsis" presents a comprehensive analysis of the ClBRN1 gene, a member of the OsNAC7 subfamily, cloned from C. lavandulifolium. However, there are certain areas where improvements could be made to enhance clarity, completeness, and readability.
Introduction: Refining the flow, clarifying the relevance to the study, and reducing redundancy will make the introduction more engaging and easier to follow, ultimately providing a stronger foundation for the manuscript. A more explicit statement of the problem or research gap early on would help to focus the reader’s attention on the importance of NAC TFs in the context of the study. Summarize the mention of specific transcription factors (e.g., SlNAP1, SlNAC10) and their effects in different plants. It might be more effective to condense some of the background details or focus on NAC TFs more selectively on the most relevant studies that directly inform the research question. There are several instances where the introduction feels disjointed, particularly when transitioning between different studies and stress types (e.g., from cold stress to salt stress). The introduction section effectively highlights the significance of NAC TFs in stress responses across various plant species, but the relevance to the current study could be emphasized more strongly. While the introduction eventually mentions the importance of these factors in chrysanthemum, this connection could be made earlier and more prominently to better contextualize the research. The literature review might benefit from a more critical evaluation of the existing literature, identifying gaps or inconsistencies that the current study aims to address. This would help to position the research within the broader scientific discourse more effectively. The justification for focusing on the ClBRN1 gene from C. morifolium and its role in stress resistance is introduced towards the end of the section. This is crucial information that could be highlighted earlier in the introduction to clearly establish the study’s significance and objectives. There are a few repetitive phrases and ideas, particularly regarding the economic importance of chrysanthemums and the impact of stress on their ornamental quality. These points could be consolidated to avoid redundancy and keep the introduction concise. The introduction ends with the statement of the study’s aims, which is appropriate. While the introduction mentions that few studies have investigated NAC transcription factors in ornamental plants, it does not specify what exact knowledge gaps this study aims to address. The introduction should explicitly state the hypothesis or research question that drives the study. It seems to provide background information but lacks a clear transition to the specific aim of the research.
Materials and Methods: Explain why the floral dip method used for Arabidopsis transformation, and why were particular vectors and plasmids selected? It would be beneficial to include subheadings within each major section to further break down the procedures. For instance, under "Target Gene Cloning," distinct subheadings for RNA Extraction, Reverse Transcription, and PCR Amplification could improve readability. However, some sections are quite dense, with technical information packed into long sentences. For instance, the description of the "Construction of the Overexpression Vector" could be broken down into more manageable parts to enhance clarity. It might be helpful to include catalog numbers for the reagents where possible. There are some areas where additional details would be useful. For example, the conditions under which Arabidopsis thaliana was grown (e.g., light intensity, photoperiod, and temperature) should be specified in the "Plant Materials and Growing Environment" section. This information is crucial as it can significantly influence the results of genetic transformation experiments. After describing the transformation of Arabidopsis, a brief statement about how this transformation relates to the subsequent stress treatments would create a smoother narrative flow. How does the "Observation of Subcellular Localization Signals" relate to the overall objective of studying the ClBRN1 gene’s role in stress response? Why was Nicotiana benthamiana chosen for the subcellular localization studies? Providing justifications that would help to underscore the validity of the methodological approach. The description of stress treatments (e.g., NaCl and cold stress) is brief. Add experimental design, replication information and including specific concentrations and duration for each treatment could enhance reproducibility. The description of paraffin sections lacks detail on how the sections were prepared, stained, and analyzed. Please specify the software used to assess the statistical significance of the data. Additionally, describe the process for generating the graphs and how the lettering was assigned to the bars. It is also important to mention the confidence interval used to determine statistical significance. I recommend including this information in a new subsection titled "Statistical Analysis" at the end of the Materials and Methods section.
In result section, the main areas for improvement include enhancing the interpretative depth, improving the narrative flow, addition of key values in terms of percentage increase or decrease and providing more context and statistical analysis. Additionally, streamlining the text for clarity and conciseness would make the results more accessible and impactful. Some descriptions, particularly in the bioinformatics analysis, could be more concise. Streamlining the text to focus on the most critical findings without losing essential information would improve readability. It would be helpful to include statistical analyses (e.g., p-values or confidence intervals) to strengthen the claims made. For example, when discussing differences in plant height or cell wall thickness, providing statistical evidence of significance would substantiate the observed differences (percentage increase or decrease). While the results are described in detail, the interpretation could be more robust. The section tends to focus on what was observed without sufficiently discussing why these observations are significant or what they imply about the function of the ClBRN1 gene. Adding a brief interpretation after each set of results would help contextualize the findings within the broader framework of the study. When discussing the NAM domain or NAC transcription factor family, a brief explanation or reference to their known functions in plants would help readers unfamiliar with these terms.
Discussion: The interpretation could delve deeper into the mechanistic aspects. How exactly might ClBRN1 confer increased salt and low-temperature tolerance at the molecular level? Providing speculative mechanisms based on the localization data and secondary cell wall involvement would add depth. The discussion could further explore why ClBRN1 might exhibit similar or different behaviors compared to these other NAC proteins. Are there unique features in ClBRN1 that account for its specific effects in Arabidopsis? This section appropriately justifies the use of C. lavandulifolium as a model for chrysanthemum studies due to its diploid nature and favorable growth characteristics (It must be added earlier in the introduction section). This context is essential for readers to understand the significance of cloning ClBRN1 from this species. Expanding on how findings in Arabidopsis might translate to Chrysanthemum would further strengthen the relevance. Please discuss how might ClBRN1 be utilized in breeding programs to develop stress-resistant chrysanthemum varieties? Discussing practical applications would enhance the impact of the research. A more comprehensive acknowledgment of the study's limitations would provide a balanced perspective. For instance, discussing the reliance on Arabidopsis as a heterologous system and any potential differences in gene function between species would be pertinent. The section could expand on specific future research directions. This section lacks detailed mechanistic insights. Delving into how ClBRN1 influences gene expression related to stress tolerance or secondary cell wall biosynthesis would enhance scientific depth. For instance, does ClBRN1 activate specific stress-responsive genes or interact with hormonal pathways like ABA signaling? The connection between ClBRN1 overexpression and increased secondary cell wall thickness is noted, aligning with the role of NAC transcription factors in secondary wall biosynthesis. However, elaborating on the significance of this finding—such as improved structural integrity or altered mechanical properties—and how it might contribute to stress tolerance would provide a more comprehensive understanding. Occasionally, the language could be streamlined for better readability. For example, sentences like "Chrysanthemum is a traditional Chinese flower with many varieties, but its growth and ornamental value are often limited under stress..." could be more concise: "Chrysanthemum, a traditional Chinese ornamental flower with numerous varieties, often experiences limited growth and ornamental value under stress conditions..." Ensure consistency in terminology, such as referring to "Arabidopsis thaliana" uniformly rather than alternating between "Arabidopsis" and "Arabidopsis thaliana." Additionally, precise language regarding the findings (e.g., specifying the degree of increased stress tolerance) would enhance clarity. Incorporating a theoretical framework or model that synthesizes the findings could aid in conceptualizing ClBRN1's multifaceted roles. This could involve proposing a model where ClBRN1 interacts with specific signaling pathways to regulate both growth and stress responses. Discussing the broader implications for plant biology, agriculture, or horticulture—beyond immediate applications in chrysanthemum breeding—would enhance the Discussion. For example, insights into NAC transcription factors like ClBRN1 could inform strategies for engineering stress-resistant crops in other species. More critical analysis of how the results fit within the context of previous research is needed. Some points in the discussion, such as the importance of NAC factors in stress responses, are repeated without adding new insights or deeper analysis.
Conclusion: The conclusion should not only summarize the findings but also discuss their implications in the context of ornamental plant breeding and stress resistance. The conclusion appears to be cut off and does not fully summarize the key findings, their implications, or the broader impact of the research. The conclusion should ideally highlight the next steps or future research directions based on the study’s findings. The conclusion should clearly articulate how the findings contribute to the existing body of knowledge and address the specific research question posed in the introduction. To enhance the conclusion, the authors should consider expanding the discussion of the study’s significance, restating the research objectives, and briefly suggesting future research directions. This will make the conclusion more robust and provide a stronger, more reflective end to the paper.
Specific comments:
Line 67-69: Zhong et al., 2006; Ko et al., 2007; Mitsuda et al., 2007; Soyano et al., 2008; Bennett et al., 2010; Yamaguchi et al., 2010; Fendrych et al., 2014; Zhou et al., 2014; Endo et al., 2015; Zhou et al., 2015). Line No. 70-71 (Mitsuda et al., 2005; Mitsuda et al., 2007; Tan et al., 2018; Lee et al., 2019; Zhong et al., 2021). Reduce the number of references.
Line No. 198-201: the fluorescence of Super1300::ClBRN1::GFP mainly appeared in the cell membrane and nucleus of epidermal cells, while the GFP signal of the control Super1300::GFP empty vector mainly appeared around the cell membrane, cytoplasm and nucleus. These results indicate that the ClBRN1 protein is localized to the cell membrane and nucleus. Explain Epidermal cells of which part? And GFP signal of the control Super1300::GFP appeared in which types of cells and in which parts?
Line 210-212: The statistical results of Arabidopsis height (Figure 3B) showed that the height of the transgenic Arabidopsis plants was lower than that of the wild-type Arabidopsis plants after 17 days. Which type of statistical analysis? And how much lower? Give key values.
Double check the references cited in the text and provided in the reference section to ensure their accuracy.
Sequences of primers and their application. Correct the caption of Table 1 and add detail, it must be self-explanatory.
Figure 2 Subcellular Localization Analysis of ClBRN1, mention subcellular parts and types of cells in figure caption.
There are significant issues with the letter assignments in Figures 3 and 5. The assignment of letters appears to be incorrect, and no mention of statistical software is provided. It is unclear how the authors assigned these letters, how the bar graphs were generated, and how the error bars were marked. Furthermore, the meaning of these letters is ambiguous. If they are intended to represent levels of statistical significance, they are inaccurately applied. It is imperative that the authors include detailed information about the post-hoc tests used for letter assignment. The authors should also perform the statistical analysis again to verify the accuracy and reliability of the assigned letters, as they are currently incorrect. Please provide a comprehensive explanation of the statistical methods, including software and criteria used, to ensure the rigor and transparency of the analysis.

Experimental design

The manuscript would benefit from a clear and detailed description of the experimental design, which is currently missing. Additionally, there is no information provided regarding the statistical analysis conducted. Including these details is crucial for the reproducibility and validation of the results. Please ensure that both the experimental design and the statistical methods are thoroughly described.

Validity of the findings

The validity of the findings is difficult to ascertain in their current form due to the absence of key quantitative values, such as percentage increases or decreases. Qualitative descriptions alone are insufficient for a thorough evaluation of the results. Additionally, there are significant issues with the presentation of Figures 3 and 5. The lettering used is incorrect and lacks clear meaning. It is essential to provide accurate quantitative data and correct the issues in the figures to ensure the reliability and clarity of the results.

---

## Round 0.2 · Minor Revisions

Dear Authors
The manuscript still needs a minor revision before publication. The authors are invited to revise the paper considering all the suggestions made by the reviewer. Please note that the requested changes are required for publication.
With Thanks

Reviewer 1 ·

Basic reporting

Necessary arrangements have been made. Suggestions have been taken into consideration. Considering that the visuality of the text will be corrected in the typesetting (such as being justified), it can be accepted.

Experimental design

There does not seem to be a problem in experimental design.

Validity of the findings

The findings are logical, rational and well argued.

Additional comments

Necessary arrangements have been made. Suggestions have been taken into consideration. Considering that the visuality of the text will be corrected in the typesetting (such as being justified), it can be accepted.

Reviewer 2 ·

Basic reporting

The authors have successfully addressed most of the concerns raised in the first revision. However, there remain some typographical and writing issues that require attention. It is recommended that the entire manuscript be carefully reviewed for these errors.
Regarding lines 244-245: "After 17 days, the height of transgenic A. thaliana began to be significantly lower than that of transgenic plants, decreasing by about 37%." This statement appears unclear, as it refers to transgenic plants in both cases. It should be clarified that the comparison is likely between wild-type and transgenic plants.
Additionally, the labeling of Figure 3 and 5 remains incorrect. In Figure 3A and 3B, representing rosette diameter and plant height, it seems that the authors have applied LSD (Least Significant Difference) separately for each data collection day. It is strongly advised that the statistical analysis for LSD be applied collectively across all data points from different days for a more accurate presentation.
It also appears that the authors have analysed temperature and salinity stress separately for LSD but presented them in the same figure, which may lead to confusion. If both stress treatments are being presented in a single figure, it is recommended to apply LSD collectively across both treatments for proper statistical comparison.
Furthermore, in Figure 5A, B, and C, it is unclear why the over-expression bars are longer yet labeled with "b" rather than "a." This discrepancy needs clarification. Figure 5D also lacks both standard error bars and letters indicating the level of significance, which should be included for completeness.
Please ensure that the full forms of all abbreviations (such as WT and OE-CIBRN1) are provided in the footnotes of the tables and figures. It is important to make the figures and tables self-explanatory for ease of understanding and clarity for the readers.

Experimental design

Experimental design seems sound.

Validity of the findings

The findings are promising; however, it is imperative to perform the statistical analysis for Figures 3 and 5 as outlined in the basic reporting. The figures should be reconstructed accordingly. This step is critical to ensure the validity and reliability of the results.

---

## Round 0.3 · accepted · Accept

Dear Authors,

I am pleased to inform you that the manuscript has improved after the last revision and can be accepted for publication.

Congratulations on accepting your manuscript, and thank you for your interest in submitting your work to PeerJ.

With Thanks

Reviewer 2 ·

Basic reporting

The authors have successfully addressed all prior suggestions. The revised manuscript now presents a well-structured and scientifically rigorous study, making a valuable contribution to the field. The manuscript is recommended for publication in PeerJ.

Experimental design

Experimental design is sound.

Validity of the findings

Findings seems valid.